# Competing electronic states emerging on polar surfaces

**Michele Reticcioli** [1,2], **Zhichang Wang**[2,3], **Michael Schmid**[2], **Dominik Wrana**[4], **Lynn A. Boatner**[5], **Ulrike Diebold** [2], **Martin Setvin**[2,4] ✉ **& Cesare Franchini** [1,6] ✉

Excess charge on polar surfaces of ionic compounds is commonly described by the two-dimensional electron gas (2DEG) model, a homogeneous distribution of charge, spatially-confined in a few atomic layers. Here, by combining scanning probe microscopy with density functional theory calculations, we show that excess charge on the polar $TaO_2$ termination of $KTaO_3$(001) forms more complex electronic states with different degrees of spatial and electronic localization: charge density waves (CDW) coexist with strongly-localized electron polarons and bipolarons. These surface electronic reconstructions, originating from the combined action of electron-lattice interaction and electronic correlation, are energetically more favorable than the 2DEG solution. They exhibit distinct spectroscopy signals and impact on the surface properties, as manifested by a local suppression of ferroelectric distortions.

Excess charge emerges spontaneously at the surfaces and interfaces of polar ionic compounds and dictates the physical and chemical material properties[1]. Usually, this intrinsic charge is described as a homogeneous two dimensional electron gas (2DEG), charge carriers spatially confined in the near-surface atomic layers[2,3] that can trigger different phase transitions[4–8] and can be functionalized for a variety of applications[9–12]. On the other hand, signatures of ferromagnetic domains and superconductivity at the interfaces of these materials suggest the possibility for inhomogeneous distributions of the interfacial excess charge[13–18]. Moreover, non-intrinsic excess charge on neutral and polar surfaces, generated by chemical or lattice defects, is known to form localized polaron states[19,20], excess holes or electrons coupled to local lattice distortions[21]. Localized polarons cause deep modification of surface properties, fundamentally different from those induced by a dispersed 2DEG, and play a key role in all applications involving charge transport and catalysis[22–25]. One question that has remained unanswered to date is whether intrinsic excess charge on polar surfaces can spontaneously form localized polaron states or charge inhomogeneities on the defect-free termination, thus providing an alternative scenario beyond the commonly accepted, delocalized 2DEG picture.

Here, by combining scanning probe microscopy experiments with density-functional theory (DFT) simulations, we examine the (001) surface of the cubic quantum-paraelectric perovskite $KTaO_3$[26]. This is a prototypical polar surface expected to host a homogeneous 2DEG on the ferroelectrically-polarized $TaO_2$ termination[27]. Our data show that the intrinsic excess charge on $KTaO_3$ is instead accommodated through the formation of charge-density waves (CDW, periodic charge modulations occurring in a variety of materials)[28–31] and highly-localized small electron polarons and bipolarons (singly and doubly reduced Ta atoms, respectively). To weaken their mutual repulsive interaction, polarons and bipolarons form ordered patterns, and are found to coexist with the CDW. The coupling between charge localization and the crystal lattice can induce significant structural distortions that disrupt the surface ferroelectric polarization.

[1]University of Vienna, Faculty of Physics, Center for Computational Materials Science, Vienna, Austria. [2]Institute of Applied Physics, Technische Universität Wien, Vienna, Austria. [3]State Key Laboratory for Physical Chemistry of Solid Surfaces, Collaborative Innovation Center of Chemistry for Energy Materials, and Department of Chemistry, College of Chemistry and Chemical Engineering, Xiamen University, Xiamen, China. [4]Department of Surface and Plasma Science, Faculty of Mathematics and Physics, Charles University, 180 00 Prague 8, Czech Republic. [5]Materials Science and Technology Division, Oak Ridge National Laboratory, Oak Ridge, TN, USA. [6]Dipartimento di Fisica e Astronomia, Università di Bologna, 40127 Bologna, Italy. ✉e-mail: setvin@iap.tuwien.ac.at; cesare.franchini@univie.ac.at

The formation of ordered patterns of (bi)polarons and a CDW on the ferroelectrically active KTaO$_3$(001) is expected to impact the functionalities of the material as compared to the simple 2DEG picture. From a more fundamental point of view, our study demonstrates much richer physics of polar surfaces than considered so far, revealing charge modulation as an intrinsic property of the bare surface. In the following, we start by showing the experimental evidence for the coexistence of different types of electronic states on the surface. We then analyze these states on the basis of first-principles DFT results.

## Results and discussion

Figure 1 shows the results of our surface-sensitive experiments on cleaved KTaO$_3$(001) samples, obtained from slightly $n$-type-doped crystals with substitutions of Ca atoms on K sites to ensure bulk electrical conductivity for the measurements (an extended data set on variously doped crystals is shown in the Supplementary Figs. 1–7). Figure 1a shows a constant-height, non-contact atomic force microscopy (ncAFM) image of the bulk-terminated surface with atomically resolved KO terraces alternating with regions of TaO$_2$[27]. (The TaO$_2$ regions lie lower by half a unit cell and are unresolved in the image.) Figure 1b–f show local conductance (LDOS) maps and scanning tunneling spectroscopy (STS) curves measured in the same region.

The occupied LDOS shows a complex structure confined on the TaO$_2$ regions, while the vicinity of the steps (i.e., borders of the TaO$_2$ terraces) as well as the KO terraces are free of in-gap states[27]. The electronic structure observed near the Fermi level (Fig. 1b) shows a wave pattern with a wavelength (peak-peak) of $1.55 \pm 0.15$ nm, typically rotated by $\approx 45 \pm 10°$ with respect to the $\langle 100 \rangle$ directions (see Supplementary Figs. 1, 2). This has been previously interpreted as a simple 2DEG[27,32,33], but detailed analysis of STS data brings a solid evidence for the presence of a charge density wave: There is a small band gap at the Fermi level (Fig. 1f), the wavelength does not disperse in energy (Supplementary Fig. 1), the empty states show a phase reversal with respect to the filled states (Supplementary Fig. 4)[31], and the waves disappear above $\approx 80$ K (Supplementary Fig. 6). At energies below $\approx -0.1$ V, where the LDOS is not dominated by the CDW, localized spots appear in the LDOS maps (Fig. 1c–e), indicating the presence of small polarons. These polaronic states exhibit a partially ordered pattern and are located in the minima of the occupied CDW states, as expected for repulsion of negative charges.

To gain insight into the experimental observations, we analyzed different electronic surface states (2DEG, CDW and (bi)polarons) individually, by performing DFT calculations modeling the TaO$_2$ termination of KTaO$_3$ with $2\sqrt{2} \times 2\sqrt{2}$-large slabs, and including an on-site interaction parameter of $U = 4$ eV to account for electronic correlation (see "Methods" section). Figure 2 highlights the essential structural and electronic features of the surface. By adopting a simple oxidation-state model, KTaO$_3$ can be considered as built by negatively (KO)$^-$ and positively (TaO$_2$)$^+$ charged layers, stacking along [001] with large interlayer charge transfer[27]. Broken bonds at the TaO$_2$ surface boundary interrupt the charge transfer leading to an accumulation of 1/2 electron per $1 \times 1$ surface unit cell (as sketched in Fig. 2a)[34]. These uncompensated electrons constitute the intrinsic excess charge of the polar surface.

The conventional homogeneous 2DEG solution is shown in Fig. 2b, c. The 2DEG spreads over the surface and shallow sub-surface TaO$_2$ layers, and it is associated with metallic states at the bottom of the conduction band (see the density of states (DOS) in Fig. 2b); conversely, deeper TaO$_2$ layers are not affected by the metallic 2DEG and retrieve the bulk-like semiconducting DOS. The top view and the simulated scanning tunneling microscopy (STM) images in Fig. 2c further highlight the spatial homogeneity of the 2DEG on the Ta sites of the $1 \times 1$ uniform surface lattice. We note that the excess electrons

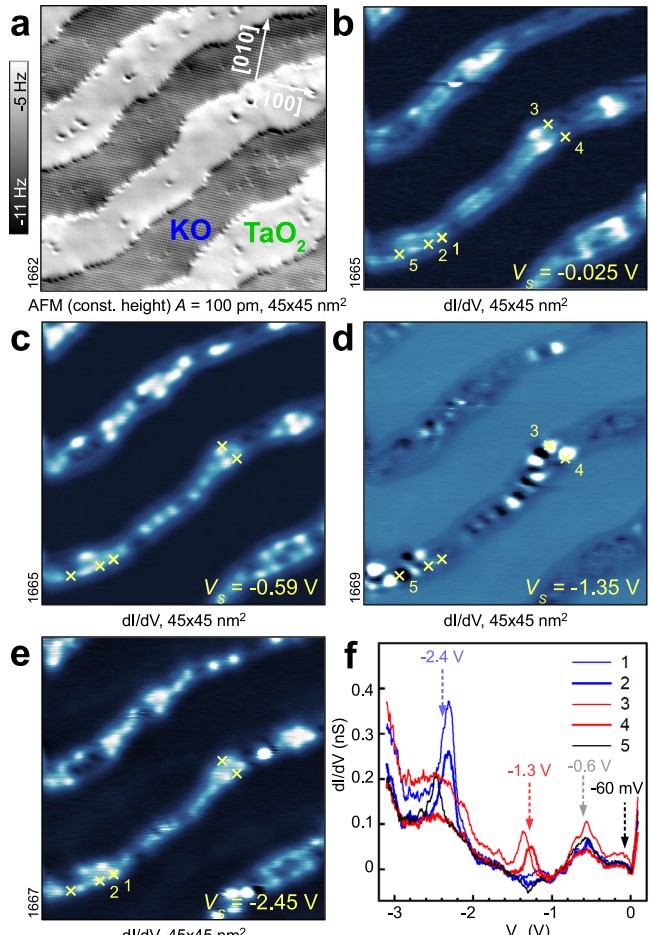

**Fig. 1 | Spatially resolved electronic structure of bulk-terminated KTaO$_3$(001).** All data acquired in constant-height mode at $T = 4.8$ K. **a** ncAFM image showing terraces with alternating KO and TaO$_2$ terminations. **b–e** Conductance maps of the same area at different sample voltages $V_S$. **f** STS spectra measured at positions marked in panels (**b–e**).

here carry parallel spin moments, forming a spin-polarized 2DEG with the conduction band of the minority spin channel completely empty (a spin-degenerate 2DEG would be 50 meV per excess electron less stable, see Supplementary Fig. 8). Importantly, the surface undergoes a ferroelectriclike (FE) distortion: While atoms on bulk KO and TaO$_2$ layers are perfectly aligned on their respective planes (as expected for a quantum paraelectric crystal as KTaO$_3$)[35], O atoms on the surface appear displaced outwards along [001] by $\delta_{FE} = 0.21$ Å from the Ta plane (see also Table 1, and Supplementary Fig. 9 for further details on the surface structure). This is a typical feature of bulk-terminated polar surfaces, as ferroelectric displacements create local dipole moments opposed to the internal electric field, in order to stabilize the surface against the diverging internal electrostatic potential and the polar catastrophe[27,36–38].

Figure 3 collects the alternative solutions (CDW and (bi)polarons), all significantly more stable than the homogeneous 2DEG (see Table 1). A spin-polarized CDW can form on the surface ($\Delta E = -50$ meV per excess electron as compared to the homogeneous 2DEG), showing a weak modulation of the charge with $\sqrt{2} \times \sqrt{2}$ periodicity, revealed by magnetic moments of 0.5 and 0.2 $\mu_B$ on Ta ions in a checkerboard-ordered pattern (Fig. 3a). We note that the formation of this CDW does not require any additional structural distortion with respect to the homogeneous 2DEG phase, preserving the FE distortions on the surface (see Table 1). The electronic fingerprint of the CDW phase consists of a shallow peak right below the Fermi level,

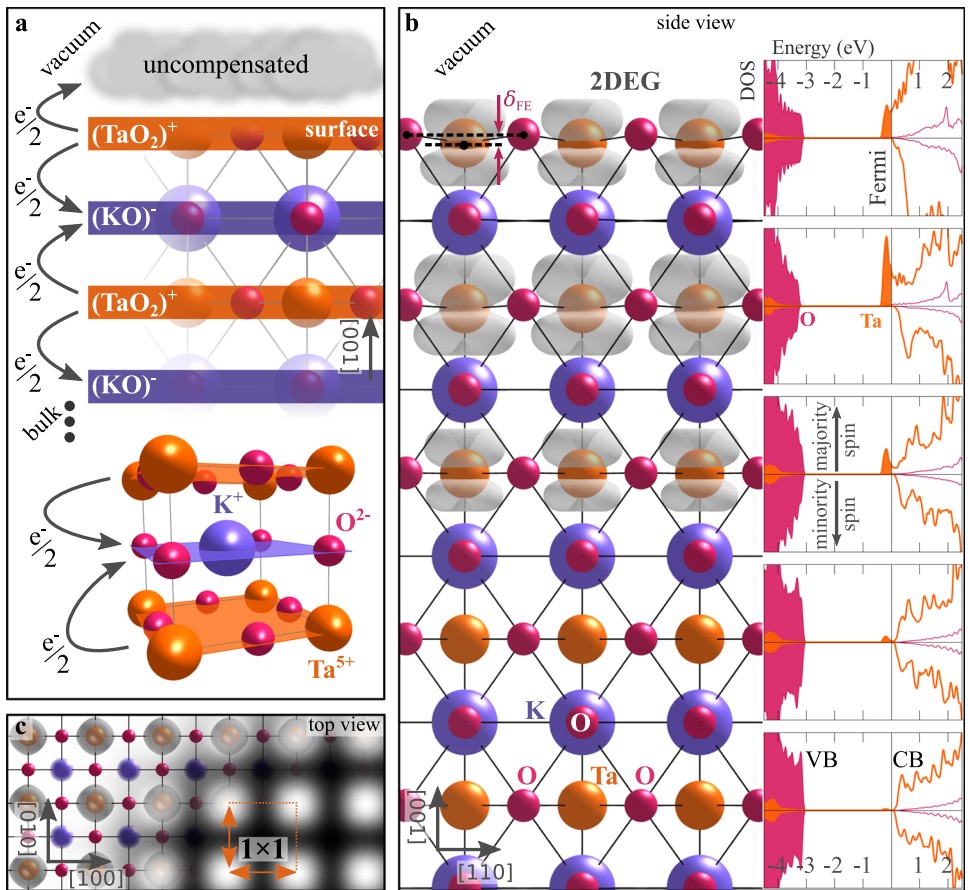

**Fig. 2 | 2DEG on the polar KTaO₃ surface. a** Pictorial view of the perovskite KTaO₃, sketching the bulk primitive cell with the formal oxidation state of atoms (bottom), and the $(KO)^-$ and $(TaO_2)^+$ layer stacking below the (001) surface (top); the arrows between layers highlight the charge transfer between consecutive layers. **b** Side view of the KTaO₃(001) hosting a spin polarized 2DEG as obtained by DFT simulations (electronic charge of the excess electrons represented in gray); the $\delta_{FE}$ label indicates the ferroelectriclike distortions, given by the O ions lying above the Ta plane; the insets show the spin-resolved DOS between conduction (CB) and valence (VB) bands, projected on the Ta (orange) and O (red) atoms separately for every TaO₂ layer (arbitrary units). **c** Top view of the system in panel (**b**), and corresponding simulated filled-state STM image, with clear 1 × 1 symmetry.

## Table 1 | Properties of the excess charge

| Excess charge state | $\Delta E$ (meV) | Magnetic moment ($\mu_B$) | Surface $\delta_{FE}$ (Å) | $\Delta b_{Ta-O}$ (Å) |
|---|---|---|---|---|
| Homegeneous 2DEG | 0 | 0.3 | 0.21 | 0 |
| $\sqrt{2} \times \sqrt{2}$ CDW | −50 | 0.2–0.5 | 0.21 | 0 |
| $\sqrt{2} \times \sqrt{2}$ polarons | −230 | 0.8 | 0.23 | +0.03 |
| [110] bipolarons | −180 | 1.6 | 0.08 | +0.09 |
| Bipolaron + polarons | −250 | | | |

Energy stability $\Delta E$ (meV per excess electron), magnetic moment of Ta atoms, ferroelectriclike distortion $\delta_{FE}$ of the surface, and changes of the Ta–O bond lengths, for the homogeneous 2DEG, CDW, single-electron polarons (in the optimal $\sqrt{2} \times \sqrt{2}$ arrangement on the sub-surface layer), and surface bipolarons aligned along [110], as well as the energy stability for the mixed configurations of polarons and bipolarons. $\Delta b_{Ta-O}$ represents the breathing distortions (averaged bond length changes) of the in-plane O atoms surrounding the (bi)polaronic Ta. For CDW charge modulation, minimum and maximum values of the local magnetic moment are shown. Additional details in the "Methods" section.

and an almost vanishing LDOS signal at $E_F$ (see DOS in Fig. 3a), in agreement with the experimental observations (Fig. 1f).

Figure 3 b,c show the most representative localized (bi)polaron solutions. The most favorable phase is formed by an ordered arrangement of bipolarons and single-electron polarons (Fig. 3b, $\Delta E = -250$ meV). Alternative (bi)polaronic solutions are shown in Fig. 3c (bipolarons on the surface aligned along the [110] direction, $\Delta E = -180$ meV), Supplementary Fig. 10 (sub-surface single-electron

polarons, $\Delta E = -230$ meV) and Supplementary Fig. 11 ([100]-ordered bipolarons, $\Delta E = -100$ meV). The bipolaronic Ta³⁺ surface sites posses a magnetic moment of 1.6 $\mu_B$, in contrast to 0.8 $\mu_B$ in surface and subsurface single-electron polaron Ta⁴⁺ sites. The (bi)polaronic states are insulating and associated to in-gap peaks (see DOS in Fig. 3b, c) similar to the peaks at deep energies observed in the experiment (Fig. 1f). The two electrons forming the bipolaron in Fig. 3c give rise to degenerate in-gap peaks, which are split in the mixed phase in Fig. 3b due to the repulsive interaction with the surrounding single-electron polarons.

Due to the particularly strong charge confinement, bipolarons affect the lattice structure considerably, causing breathing-out displacements of oxygen atoms surrounding the Ta³⁺ site (Ta–O bond length increased by $\Delta b_{Ta-O} = +0.09$ Å), and a local quenching of the ferroelectric distortions ($\delta_{FE} = 0.08$ Å). Single-electron polarons instead induce smaller lattice distortions ($\Delta b_{Ta-O} = +0.03$ Å) and preserve the surface ferroelectricity ($\delta_{FE} = 0.23$ and 0.15 Å for subsurface and surface polarons, respectively).

In order to acquire a detailed understanding of the various states, we show in Fig. 4 the corresponding band structures. In the 2DEG phase the $t_{2g}$ bands ($d_{xy}$, $d_{xz}$, and $d_{yz}$) cross the Fermi level in the majority spin channel, whereas the corresponding minority-spin states are unoccupied (Fig. 4a). The observed band splitting is mainly caused by the internal electric field of the polar surface[32,39]. In the CDW phase, the charge modulation is associated to low-dispersion $d_{xz}$ and

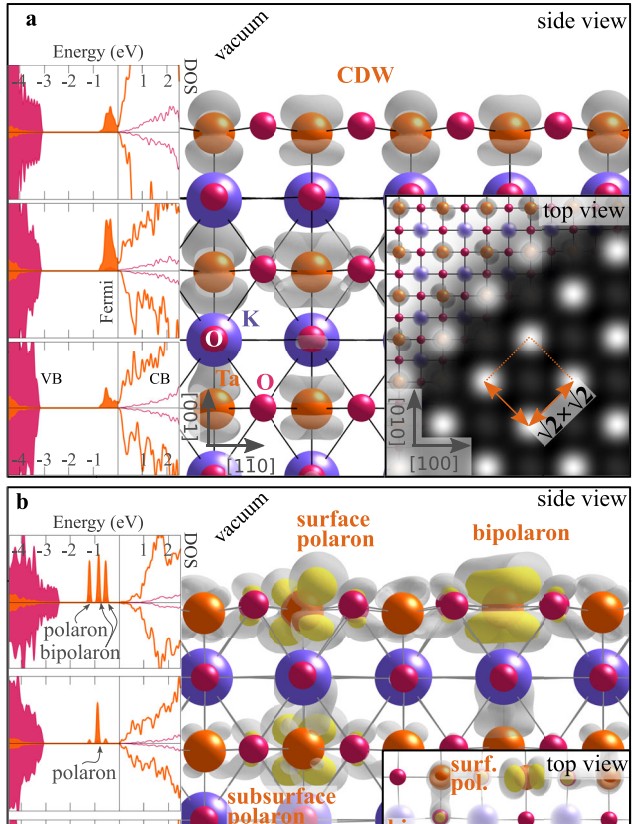

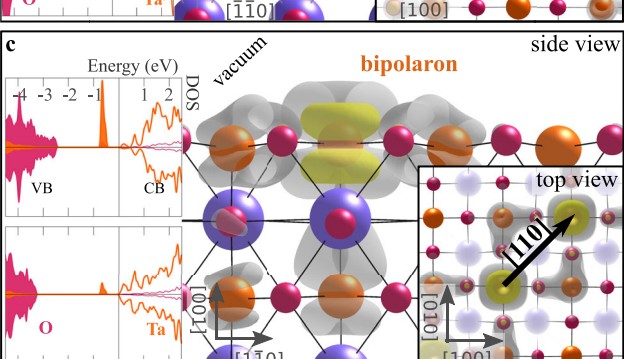

**Fig. 3 | Charge density wave and (bi)polarons. a** Side view of the CDW (electronic charge represented by gray cloud); insets show the spin-resolved DOS projected on the TaO$_2$ layers (arbitrary units), the top view, and the simulated filled-state STM signal. Side view, top view, and DOS for the mixed configuration of single-electron polarons and bipolarons (**b**) and the bipolaron [110] arrangement (**c**). Yellow and gray colors represent the polaronic charge density at high and low isosurface levels, respectively.

$d_{yz}$ bands lying below the Fermi level, while the $d_{xy}$ bands are completely empty (Fig. 4b). The strong charge localization and the coupling with lattice distortions associated to the formation of polarons (Fig. 4c) and bipolarons (Fig. 4d) tends to form essentially dispersionless in-gap bands: deviations from completely flat bands reveal the repulsive interactions perturbing the (bi)polaronic states[40].

Summing up at this point, we found that the uncompensated surface charge can be modeled with three different electronic phases: 2DEG, CDW, and (bi)polarons. The stability of these phases is controlled by the degree of charge ordering (facilitated by electron

localization associated with sizable electron correlation) and local lattice distortions (Ta–O bond length and ferroelectriclike distortions), mainly originating from electron-phonon interactions and phonon instabilities. By removing the on-site $U$, charge localization is suppressed and the homogeneous 2DEG becomes the dominant phase (see Supplementary Fig. 12)[32,33], in disagreement with our experimental observations. To decode the effect of structural distortions, we performed a series of calculations from the uniform 2DEG limit to the distorted mixed polaronic solution (i.e., the global energy minimum configuration shown in Fig. 3b) by interpolating the ion coordinates between these two end states (Fig. 5): In the FE-distorted surface (in the absence of polaronic-like oxygen breathing distortions), the CDW phase is more stable than the homogeneous 2DEG. By switching on polaron breathing displacements and modeling the evolution towards the fully bipolaron state, the system passes through an intermediate phase characterized by the coexistence of CDW and single-electron polarons (Fig. 5a). The transition from the bipolaron to the mixed polarons+bipolaron state is achieved by splitting one bipolaron into two single-electron polarons, as shown in the insets of Fig. 5 and corresponding energy profile. The relative stability of one phase over the other is controlled by the degree of lattice distortion that plays the role of the order-parameter in analogy to similar quantum-critical ferroelectric and multiferroics transitions[26,41].

Our computational data show that polaronic states can coexist with the CDW, in agreement with the experiment. We note that a precise characterization of the relative stability of the electronic surface states is hampered in the DFT calculations by the dependence of the free energy on the computational setup (see Supplementary Fig. 12)[42]; moreover, the infinitely large TaO$_2$ terminations in the calculations neglect the effects of the surface electrostatic field arising from the neighboring KO terraces, known to alter the surface electronic states[27]; also, the limited size of the DFT unit cell constrains the electronic phases, the CDW in particular. Nevertheless, the small differences in the formation energy of the different phases (−50 to −250 meV), including the CDW+polaron phase, support the possibility of a coexistence of these states as observed on the experimental samples. In fact, the DOS in Fig. 3 and band structures in Fig. 4 allow for interpretation of the experimental STS results from Fig. 1. The STS shoulder at −60 mV is at the bottom of the conduction band, separated from the empty states by a small gap induced by the CDW. The simulated CDW exhibits a wavelength of about 5.7 Å (see Fig. 3a), increased to 11.4 Å in the mixed CDW+polaron phase (see Supplementary Fig. 13), slightly shorter than the experimental one (15.5 ± 1.5 Å; a precise modeling of the experimentally observed CDW would require a prohibitively large supercell size). The peaks measured at −0.6 and ≈−1.3 V can be attributed to polaron or bipolaron states. While LDOS measurements do not allow us for unambiguous assignment of these deep states with specific polaronic types, support for predominance of Ta$^{3+}$ bipolarons in KTaO$_3$ over Ta$^{4+}$ single-electron polarons can be found for instance in photoluminiscence experiments[43,44]. Further STS maxima at more negative bias depend on the tip; possibly these (as well as the negative tunneling conductance sometimes observed) are related to changes of the charge configuration by the electric field of the tip.

In summary, we have shown the emergence of coexisting charge density waves, polarons, and bipolarons originating from the intrinsic uncompensated charge of the polar TaO$_2$-terminated KTaO$_3$(001) surface. We have obtained a direct, real-space view of these states, in contrast to similar phenomena at interfaces of heterostructured materials[17,45–47], where the charge distribution is not directly accessible. The competition between charge-ordering (CDW and (bi)polaronic states) and homogeneous distribution (2DEG) arises from the contrasting action of spatially extended 5$d$ orbitals of Ta (favoring charge delocalization) and strong electronic correlation (favoring

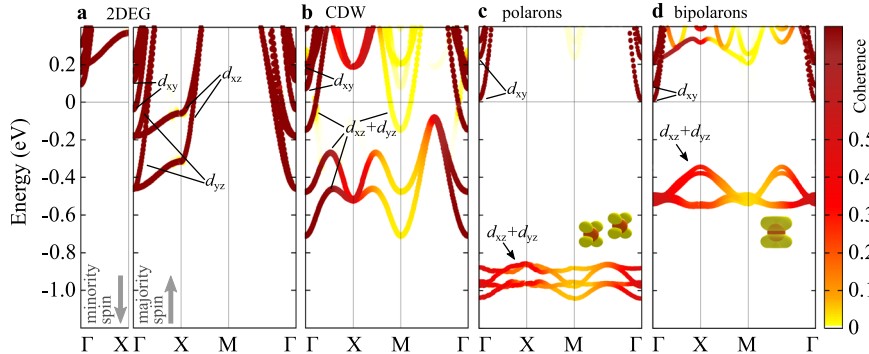

**Fig. 4 | Energy band structures.** Energy bands of the TaO₂ surface as obtained by modeling separately the 2DEG (**a**), CDW (**b**), sub-surface single-electron polarons (**c**), and bipolarons aligned along [110] (**d**). The color gradient from white to brown indicates the coherence of the supercell states unfolded in the Brillouin zone of the primitive cell (see "Methods" for details on the unfolding procedure). For the 2DEG (**a**) both spin channels are shown. The Fermi level is arbitrarily set at the bottom of the conduction band for the insulating cases (**c**, **d**). Labels indicate the $d_{xy}$, $d_{xz}$, and $d_{yz}$ dominant orbital characters.

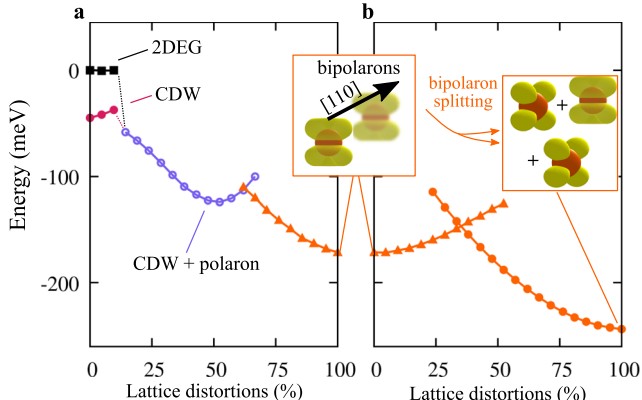

**Fig. 5 | Effect of local lattice distortions. a** Energy dependence of all electronic states on the lattice distortions; the ions are progressively moved (by using a linear interpolation, see Methods Section) from the FE-distorted 1 × 1 lattice (0%) to the structure distorted by bipolarons aligned along [110] (100%); a transient phase is observed, showing single-electron polarons coexisting with the CDW (with a local energy minimum around 50%). **b** Bipolaron splitting, i.e., transition from bipolarons aligned along [110] (0%) to a mixed configuration of bipolarons and surface and sub-surface polarons (100%), representing the most stable charge ordering pattern.

charge localization). The entanglement of the electronic degree of freedom with a phonon-active lattice enabled by electron-phonon coupling is crucial for the stabilization of charge trapping in the form of (bi)polarons, associated with breathing-out displacements of oxygen atoms surrounding Ta. The charge ordering originating from the large amount of uncompensated charge available on the polar surface is an interesting feature, which might explain the long lifetimes of polarized microdomains induced by photocarriers[43,48]. The possibility to control surface ferroelectric polarization by charge trapping through the formation of bipolarons represents a novel physical effect that could be important for applications such as piezocatalysis or pyrocatalysis[49], where the surfaces turns polar once the bulk material switches between the ferroelectric and paraelectric states.

## Methods

### Computational methods

DFT calculations were performed by using the Vienna ab initio simulation package (VASP)[50–52]. We adopted the strongly constrained and appropriately normed meta-generalized gradient approximation (SCAN)[53], with the inclusion of an on-site effective $U$[54] of 4.0 eV on the $d$ orbitals of Ta atoms (see Supplementary Fig. 12). The TaO₂ termination of KTaO₃(001) was modeled by using a slab with six TaO₂ layers alternating with five KO layers, in a symmetric setup (mirror symmetry on the central (001) KO layer), including a vacuum region of more than 30 Å. To study the penetration of the 2DEG into deep layers, we used slabs with up to ten and nine TaO₂ and KO layers, respectively. In order to enable spatial symmetry breaking we used $2\sqrt{2} \times 2\sqrt{2}$ large slabs[55,56]. All the atomic coordinates (except those on the central KO layer) were relaxed using standard convergence criteria (residual forces smaller than 0.01 eV/Å), with a plane-wave energy cutoff of 500 eV, and a 3 × 3 × 1 grid for the integration in the reciprocal space, while electronic self consistence and densities of states were calculated by using a finer 7 × 7 × 1 grid. The various solutions for the intrinsic, uncompensated charge were obtained by adopting different initial conditions for the electronic density and wavefunction in unconstrained self-consistent calculations (see also Supplementary Information). We used VESTA[57] to show the spatial extension of the excess charge and the Tersoff–Hamann approximation[58] for the STM simulations, including all in-gap states and states at the bottom of the conduction band up to the Fermi energy level. In order to analyze the band structure of our large unit cells, we adopted the unfolding technique as implemented in ref. [40], by calculating the coherence (or Bloch character) of the supercell eigenstates unfolded in the Brillouin zone of a 1 × 1 surface slab.

The stability $\Delta E$ of charge density waves and (bi)polarons was calculated by comparing the total free energy $E$ with the total free energy $E_{2DEG}$ obtained for the reference spin-polarized 2DEG: $\Delta E = (E - E_{2DEG})/8$. The factor 1/8 scales the energy to the $\sqrt{2} \times \sqrt{2}$ surface cell that contains one excess electron (note that our $2\sqrt{2} \times 2\sqrt{2}$ slab contains 8 intrinsic excess electrons, i.e., 4 excess electrons per side of the symmetric slab); in the specific case of single-electron polaronic states, $\Delta E$ is equivalent to the polaron formation energy $E_{POL}$[21].

### Experimental methods

Combined STM/AFM measurements were performed at a temperature $T$ of 4.8 K in a UHV chamber with a base pressure of $10^{-9}$ Pa, equipped with a commercial ScientaOmicron q-Plus LT head. Tuning-fork-based AFM sensors with a separate wire for the tunneling current ($k$ = 1900 N/m, $f_0$ = 30,500 Hz, $Q \simeq$ 30,000)[59], as well as a custom-design cryogenic preamplifier[60], were used for the AFM measurements. Electrochemically etched W tips were glued to the tuning fork and cleaned in situ by field emission and self-sputtering and treated on a Cu(001) surface to ensure a metallic character of the tip. STS spectra were measured at open-loop conditions using a lock-in amplifier, with

a modulation frequency of 123 Hz and an amplitude of 10 mV. The $KTaO_3$ samples were prepared by solidification from a nonstoichiometric melt. Samples with 0.2% Ca doping were used, which ensures enough bulk electrical conductivity for performing the STS measurements. The cleaving was performed in situ by a tungsten carbide blade in a temperature range between 250 and 300 K, as described in ref. 27.

## Data availability
Data generated or analyzed during this study are included in this published article (and its supplementary information files).

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

## Acknowledgements

This work was supported by the Austrian Science Fund (FWF) projects P32148-N36 (SUPER, funding received by C.F. and M. Se) and SFB TACO (F81, by U.D. and C.F.), and by the joint Austrian (BMBWF CZ15/2021) Czech (MSMT 8J21AT004) project (by M.R. and M. Se). Martin Setvin acknowledges the support of the Czech Science Foundation (GACR 20-21727X and GAUK Primus/20/SCI/009). Calculations were performed at the Vienna Scientific Cluster.

## Author contributions

M. Se and C.F. designed the project. M.R. and Z.W. contributed equally to this work. M.R. performed the calculations, Z.W. conducted the experiments. All authors (M.R, Z.W., M. Sc, D.W., L.B., U.D., M. Se, and C.F.) contributed to a substantial analysis and discussion of the results, and to the manuscript writing.

## Competing interests

The authors declare no competing interests.
