## [Peer Review File · Nature Communications]

Competing electronic states emerging on polar surfacesREVIEWER COMMENTS

Reviewer #1 (Remarks to the Author):

The manuscript by Reticcioli M. et al. reports a scanning probe microscopy combined with density functional theory study of the surface electronic structure of the TaO₂ terminated surface of KTaO₃(001) single-crystal samples, with the samples doped with calcium to ensure sufficient conductivity for scanning tunneling spectroscopy (STS) measurements.

The authors observed in their AFM/STS experiments some electronic features on the polar TaO₂ terminated surface. With the aid of DFT+U calculations, they interpreted the observed features to be 'charge-density waves' ('CDW') that co-exist with strongly localized electron polarons and bi-polarons at the surface, formed from the combined action of electron-phonon coupling and electronic correlation.

The experimental data itself look interesting; however, I do not see their interpretation of the data as 'CDW' co-existing with polarons very convincing. Also, there are a number of points in both the manuscript and SI that need to be addressed, see below:

1. The authors showed in fig. 1f the STS data recorded at filled-state energies ($V_s < 0$) only. Can the authors also provide data recorded in the empty-state energies ($V_s > 0$)?
2. The authors showed in fig. sf4 two $g(V)$ map slices, one recorded at $V_s > 0$ and another at $V_s < 0$. They claimed that there exists inversion in the $|g(V)|$ signals between the two, serving as STM/STS evidence of CDWs. I do not see any signal inversion between the two (maybe the inversion is too weak to be seen? I am not sure). Also, if signal inversion only exists in some very small parts of the whole TaO₂ island, I'd rather doubt the validity of their claim about the emergence of CDWs within the TaO₂ islands. Better visualization of the signal inversion to support their claim is required.
3. The authors showed figure sf5 a series of real-space $g(V)$ map slices, recorded at the energies of the 'polaron' (if they really are) peaks shown in the STS data. Based on their observations that 'polarons' only exist at positions away from point defects, the authors claimed that "the polaron positions are not correlated with structural defects in the TaO₂ termination." I don't think this is a correct statement: if there were no correlation, the 'polarons' should exist in any position of the TaO₂ termination.
4. In the $g(V)$ map slice at $V_s = -25$ mV shown in fig. 1b, the 'CDW' contrast is stripe-like, with 'stripes' running parallel to the island edges; while in that shown in fig. sf1, the contrast is checkerboard-like. Could the authors provide an explanation on such difference?
5. Further from comment 4, I do not think their experimental evidence(s) for supporting their claim about the CDWs is/are very solid. This is because, in order to claim a CDW to exist, one needs to show that the 'CDW' disappears at some higher temperature. Also, the checkerboard contrast shown in fig. sf1 might not be due to the formation of a CDW; it might arise due to the presence of 'polarons', which locally alters the surface electronic states around the 'polarons'.
6. The authors reported that the formed CDWs have measured wavelength of ~15.5 Angstroms, a value three times larger than that of CDWs if not coupled to any polarons (5.07 Angstroms), as predicted using DFT calculations. Is it possible to tell in experiment, if the wavelength of the CDWs would become shortened at higher temperature at which polarons are free to move?
7. The CDW only exist at the central part of the TaO₂ islands, but not near the island edges. Can the authors comment on this?

Overall, the manuscript requires major revision before it might get published in Nature Communications.

Reviewer #2 (Remarks to the Author):

The authors in the current work observed via scanning probe microscopy on KTaO₃(001) surface a mixed nature of the charge-density wave nearby the Fermi level, polarons and bipolarons 1 eV below the Fermi level, both coexisting with the 2-dimensional electron gas on such polar surface. The authors then used the density functional theory to calculate the lattice and electronic structure of the

CDW and polaronic states. I would suggest that this work needs some major modifications before being considered to be published on Nature Communications.

1. It has been known for decades that a spin-polarized electron liquid (e.g., the 2DEG on the polar surface shown in the current work) is unstable against the formation of a charge density wave. The theoretical works have been done by numerous groups including H. Fukuyama et al [PRB 19, 5211 (1979)], R. Gerhardt [PRB 24 1339 (1981)], A. Koulakov et al [PRL 76(3) 499 (1996)], while the experimental results that directly observed the charge ordering on electronic-liquid surfaces have been shown many times by e.g. J Hoffman et al [Science 297.5584 (2002): 1148-1151], S Kivelson et al [Reviews of Modern Physics 75.4 (2003): 1201]. The authors should demonstrate the significance of the findings in the current work, and what makes it different than the previous literature.

2. In order to persuade readers that one should consider the CDW or polaronic effects, e.g., in carrier transport applications, the authors should first offer one or a few quantitative analysis on experimental observations to show if there are nonnegligible effects (if exist) that CDW and polaronic states have on the 2D electronic gas behaviors.

3. The authors discussed the STM observations of CDW state and polaronic state in a clear way, however the DFT simulation results do not agree with the observations. The CDW DOS, according to the STM results, are close to the Fermi level, while in all simulated structures shown in the current work, only the pure CDW structure (ferromagnetic Fig.2a and antiferromagnetic Fig.SF6a), the bipolaron structure shown in Fig.SF9b, and the mixture structure of CDW+polaron shown in Fig.SF10b have CDW DOS nearby the Fermi level, all of the three having significant higher total energies (>0.1 eV) than the global ground state (the one shown in Fig.3b) and hence could become unstable. The ground state, on the other hand, does not host any CDW state nearby the Fermi level. Before discussing the CDW and polaronic effects based on the DFT results, the authors should first solve these disagreements.

4. The authors should offer enough details about how they achieved the many structures with energies between the pure 2DEG and the ground states. Are they the local minima DFT found, or from constrained-DFT optimizations?

5. The authors used the SCAN+U method with $U = 4$ eV on Ta d-orbital. The first question is how this U value is determined. Is it from constraint RPA, or linear response, or an arbitrary value? For example, the linear-response method gives smaller $U=2.27$ eV for bulk Ta in GGA+U method, while SCAN often requires a smaller U value than GGA. The authors mentioned that the CDW or polaronic state will no longer be the ground state when $U=0$. Does it mean that the conclusions of the current work, at least the theoretical part, are U-dependent and might not survive under another arbitrary U value?

Reviewer #3 (Remarks to the Author):

I have read with great interest the manuscript titled "Competing electronic states emerging on polar surfaces" from Reticcioli et al that discusses charge excesses at polar surfaces of oxides. While one usually assumes that a "simple" 2D electron gas forms at the surface in order to cancel/accommodate the polar catastrophe and/or the electric field at the interface with vacuum, authors here show that polaronic and bipolaronic states can form at the surface. Results are supported by first-principles simulations as well as scanning probe microscopy.

Firstly, the paper is very well written, the problem is accurately introduced and the whole story is accessible for a broad audience. Secondly, the physics and results discussed in the manuscript are important as authors show that instead of being delocalized over few unit cells below the surface (typically 1 to 3 uc) and form a 2D gas, the excess charge can be trapped and form polaronic states. To the best of my knowledge, this is a largely ignored scenario and it is thus a breakthrough as these results hint at new paradigms for the physics of surfaces.

Despite of the few comments on the manuscript appended below that have to be addressed, I strongly support publication of the manuscript in Nature Communications.

Being more an expert of first-principles simulations, I have few questions regarding the procedures and part of the results:

- How do the authors reach the solution with trapped electrons? Do they perform an initial calculations with a pattern of specific orbital occupancies and then let the solver relax the total energy and forces in a second step?

- Authors use the SCAN functional that was previously shown to be appropriate for ABO₃ oxides with a 3d element (PRB 100 035119). Nevertheless, they used SCAN with a +U potential on Ta d states. What is the reason for that choice? I would expect that delocalization errors are smaller for 5d elements than for 3d elements. Is it just a delocalization error appearing at the surface?

- Authors write that when removing the +U in the simulations, the polaronic states disappear. It goes along with the previous point: is it really a physical effect or a problem from the SCAN functional ? I guess another parameter free simulations such as HSE06 on the smallest slab possible could provide a full justification of this point.

- Authors observe spin polarized 2DEG at the surface. Do they have an idea of the origin of such a spin polarized gas? Have they checked if this could be antiferromagnetically ordered?

“Competing electronic states emerging on polar surfaces” ___Reply to the Reviewers

Dear Reviewers,

We acknowledge receipt of the reports on our manuscript “Competing electronic states emerging on polar surfaces” and thank all Referees for the very constructive comments.

We are glad that the First Reviewer finds our results interesting, and we share the enthusiasm of the Third Reviewer who highlights the high novelty of our work in her/his comment “this is a largely ignored scenario and it is, thus, a breakthrough, as these results hint at new paradigms for the physics of surfaces”.

We believe that the revised version of our manuscript successfully answers to all criticisms and suggestions received in the reports. In particular, we have performed additional experiments to validate our interpretation of the charge density wave, following the comments from the First Reviewer. We have also clarified the significance of our work for the study of the properties and functionalities of electronic states on polar surface as asked by the Second Reviewer. Finally, the discussion on the details of the adopted computational procedure has been largely extended, following the comments of both the Second and Third Reviewers.

Best Regards,

C. Franchini, M. Setvin, on behalf of all authors

Summary of changes

- We modified the main text and the reference list according to the suggestions received from the Reviewers.
- Dominik Wrana contributed to the additional experiments performed in light of the Reviewers’ comment, thus, he is now included as a co-author.
- The Supplementary Materials was extended by including additional Figures and Sections.

Reply to the First Reviewer

Reviewer: The manuscript by Reticcioli M. et al. reports a scanning probe microscopy combined with density functional theory study of the surface electronic structure of the TaO2 terminated surface of KTaO3(001) single-crystal samples, with the samples doped with calcium to ensure sufficient conductivity for scanning tunneling spectroscopy (STS) measurements. The authors observed in their AFM/STS experiments some electronic features on the polar TaO2 terminated surface. With the aid of DFT+U calculations, they interpreted the observed features to be ‘charge-density waves’ (‘CDW’) that co-exist with strongly localized electron polarons and bi-polarons at the surface, formed from the combined action of electron-phonon coupling and electronic correlation. The experimental data itself look interesting; however, I do not see their interpretation of the data as ‘CDW’ co-existing with polarons very convincing. Also, there are a number of points in both the manuscript and SI that need to be addressed, see below:

Authors: We thank the Reviewer for her/his careful reading of our work. Motivated by her/his comments, we have performed additional experiments in order to make our interpretations more robust, as specified in the point-by-point response below.

Reviewer: 1. The authors showed in fig. 1f the STS data recorded at filled-state energies ($V_s < 0$) only. Can the authors also provide data recorded in the empty-state energies ($V_s >$

0)?

Authors: We agree that these data should be included. We have added Fig. SF5 to the Supplementary Information.

Reviewer: 2. The authors showed in fig. sf4 two $g(V)$ map slices, one recorded at $V_g > 0$ and another at $V_g < 0$. They claimed that there exists inversion in the $g(V)$ signals between the two, serving as STM/STS evidence of CDWs. I do not see any signal inversion between the two (maybe the inversion is too weak to be seen? I am not sure). Also, if signal inversion only exists in some very small parts of the whole TaO₂ island, I'd rather doubt the validity of their claim about the emergence of CDWs within the TaO₂ islands. Better visualization of the signal inversion to support their claim is required.

Authors: We have repeated the experiments and acquired better-quality data. The new results (Fig. SF4) show the phase inversion in the majority of locations. The maxima in the occupied states are marked by dots and lines.

The phase inversion develops better in larger regions. The inversion is mostly absent in narrow TaO₂ terraces, where it seems that we mainly observe standing waves (quantum well states). This is now discussed in the text related to Fig. SF4.

Reviewer: 3. The authors showed figure sf5 a series of real-space $g(V)$ map slices, recorded at the energies of the 'polaron' (if they really are) peaks shown in the STS data. Based on their observations that 'polarons' only exist at positions away from point defects, the authors claimed that "the polaron positions are not correlated with structural defects in the TaO₂ termination." I don't think this is a correct statement: if there were no correlation, the 'polarons' should exist in any position of the TaO₂ termination.

Authors: We have corrected our statement in the Figure caption:

Data identical to Fig. 1, with positions of structural defects marked by crosses. The maps of LDOS in (b-e) show that the polaron positions are not bound to structural defects in the TaO₂ termination: The correlation is negative, i.e., polarons avoid the defects.

Reviewer: 4. In the $g(V)$ map slice at $V_g = \square 25mV$ shown in fig. 1b, the 'CDW' contrast is stripe-like, with 'stripes' running parallel to the island edges; while in that shown in fig. sf1, the contrast is checkerboard-like. Could the authors provide an explanation on such difference?

Authors: Our understanding is that there are waves in two equivalent directions: [110] and [-110]. These waves may have different amplitudes. In narrow regions, the amplitude in one direction is enhanced by interferences and scattering on the step edges. However, the perpendicular direction still contains wave with a smaller amplitude. This is visible in the new Fig. SF4.

Reviewer: 5. Further from comment 4, I do not think their experimental evidence(s) for supporting their claim about the CDWs is/are very solid. This is because, in order to claim a CDW to exist, one needs to show that the 'CDW' disappears at some higher temperature. Also, the checkerboard contrast shown in fig. sf1 might not be due to the formation of a CDW; it might arise due to the presence of 'polarons', which locally alters the surface electronic states around the 'polarons'.

Authors: The CDW state indeed disappears above r' 80 K. This is now shown in Fig. SF6. In the main text, we now mention this:

There is a small band gap at the Fermi level (Fig. 1f), the wavelength does not disperse in energy (Fig. SF1), the empty states show a phase reversal with respect to the filled states (Fig. SF4), and the waves disappear above ≈ 80 K (Fig. SF6).

Further, the stability at elevated temperatures is discussed in the Supplement:

An important characteristic of a CDW is that the stability is limited to a certain temperature range. This is shown in Fig. SF6. The states directly below and above the Fermi level are shown, since they are characteristic for the CDW. The intensity of the states near the Fermi level decreases with the increasing temperature, and these states fade away above ≈ 80 K. At $T = 51$ K, we can still recognize the wave pattern, but it is unclear whether it originates from standing waves or the CDW. We do not have a solid evidence about the phase inversion at elevated temperatures.

The scenario of disappearance of the states close to the Fermi level is unconventional: Instead of a transition to a homogeneously distributed LDOS, the electronic states seem to disappear completely. This is apparent when comparing the magnitude of the tunneling current in experiments at different temperatures. We note that all experiments were performed with a copper-functionalized tip and the images were scanned at comparable height (estimated from the AFM images). The tunneling currents become dramatically lower at elevated temperatures and also the bottom of the conduction band shifts to slightly higher energies. This indicates that the electrons arising from the polar catastrophe may become unavailable at elevated temperature, which is likely linked to the destabilization of the CDW.

Reviewer: 6. The authors reported that the formed CDWs have measured wavelength of 15.5 Angstroms, a value three times larger than that of CDWs if not coupled to any polarons (5.07 Angstroms), as predicted using DFT calculations. Is it possible to tell in experiment, if the wavelength of the CDWs would become shortened at higher temperature at which polarons are free to move?

Authors: The images at higher temperatures in Fig. SF6 do not show the CDW as clear as those at r' 5 K, therefore we can neither confirm nor exclude a temperature dependence of the CDW wavelength. We can only exclude an extremely strong temperature dependence.

We would like to mention that, according to our analysis, the experimental data show the coexistence of both polarons and CDW in the same phase. Therefore, a fairer comparison between DFT and experiment is represented by the polaron+CDW modeled in the calculations, which shows a wavelength closer to the experimental value. Unfortunately, simulating the CDW corresponding to the periodicity observed in the experiment is computationally prohibitive. We are confident that increasing the cell size to the periodicity observed in experiments, when possible, would result in stronger stability of the CDW. In the main text, we have expanded the description of the disagreement between the calculations and the experimental observations regarding the wavelength and the CDW stability:

We note that a precise characterization of the relative stability of the electronic surface states is hampered in the DFT calculations by the dependence of the free energy on the computational setup (see Fig. SF12) [42]; moreover, the infinitely large TaO₂ terminations in the calculations neglect the effects of the surface electrostatic field arising from the alternating KO terraces, known to alter the surface electronic states [27], and the limited size of the unit cell constrains the electronic phases, the CDW in particular. [...] The simulated CDW exhibits a wavelength of about 5.7 Å (see Fig. 3a), increased to 11.4 Å in the mixed CDW+polaron phase (see Fig. SF13), slightly shorter than the experimental one (15.5 ± 1.5 Å; a precise modeling of the experimentally observed CDW would require a prohibitively large supercell size).

Reviewer: 7. The CDW only exist at the central part of the TaO₂ islands, but not near the island edges. Can the authors comment on this?

Authors: The K⁺O⁻² and TaO₂ terraces create a periodic potential for electrons: K⁺O⁻² are repulsive, while Ta⁵⁺O⁻² are attractive for electrons. This potential well is filled by itinerant electrons up to a certain level that is limited by the polar catastrophe. The new image of the empty states (Fig. SF5) illustrates how the potential well expands above the Fermi level; the states above the Fermi level reach increasingly further towards the step edges.

This results was already reported in our previous work on KTaO₃ (Ref. 27). We have now modified the text to report this result more clearly:

The occupied LDOS shows a complex structure confined on the TaO₂ regions, while the vicinity of the steps (i.e., borders of the TaO₂ terraces) as well as the KO terraces are free of in-gap states [27].

Reviewer: Overall, the manuscript requires major revision before it might get published in Nature Communications.

Authors: We are confident that the new version of the manuscript meets all suggestions from the Reviewer and solves his/her concerns.

Reply to the Second Reviewer

Reviewer: The authors in the current work observed via scanning probe microscopy on KTaO₃ (001) surface a mixed nature of the charge-density wave nearby the Fermi level, polarons and bipolarons 1 eV below the Fermi level, both coexisting with the 2-dimensional electron gas on such polar surface. The authors then used the density functional theory to calculate the lattice and electronic structure of the CDW and polaronic states. I would suggest that this work needs some major modifications before being considered to be published on Nature Communications.

Authors: We indeed reported the formation of non-trivial electronic states on the KTaO₃ (001) polar surface originating from the intrinsic excess charge, in contradiction to the common expectation of a homogeneous 2DEG. We thank the Reviewer for her/his comments that have helped us to clarify several aspects of our manuscript.

Reviewer: 1. It has been known for decades that a spin-polarized electron liquid (e.g., the 2DEG on the polar surface shown in the current work) is unstable against the formation of a charge density wave. The theoretical works have been done by numerous groups including H. Fukuyama et al [PRB 19, 5211 (1979)], R. Gerhardt [PRB 24 1339 (1981)], A. Koulakov et al [PRL 76(3) 499 (1996)], while the experimental results that directly observed the charge ordering on electronic-liquid surfaces have been shown many times by e.g. J Hoffman et al [Science 297.5584 (2002): 1148-1151], S Kivelson et al [Reviews of Modern Physics 75.4 (2003): 1201]. The authors should demonstrate the significance of the findings in the current work, and what makes it different than the previous literature.

Authors: Despite the rich literature about CDW on metals and superconductors, often driven by doping and/or external magnetic field in non-1-dimensional systems, there is no clear, direct observation of CDW forming spontaneously on surfaces of polar materials arising from the intrinsic excess charge. In fact, the commonly accepted picture for the intrinsic excess charge is the homogeneous 2DEG, which, however, has led to discrepancies between theory and experiments (for example, the energy bands calculated by DFT in the homogeneous 2DEG picture fail to capture the most interesting aspects of ARPES measurements in KTaO₃, as reported by Santander-Syro et al. in Ref. 33). Recent works have reported indications for charge inhomogeneity, considered the cause of coexisting electronic phases such as ferromagnetism and superconductivity (see new references

listed below); however, these studies have been performed at the interface of polar materials, which prevents a direct observation of the charge modulations. This is contrast with our $\text{KTaO}_3(001)$ surface, where CDW (together with strongly localized polarons) is clearly accessible.

We agree with the Reviewer that this aspect was not properly described in the original version of the manuscript. Therefore, we have modified the Introduction including the following comment in order to better position our work in the context of the existing literature:

[Unchanged:] Usually, this intrinsic charge is described as a homogeneous two dimensional electron gas (2DEG), charge carriers spatially confined in the near-surface atomic layers [2,3] that can trigger different phase transitions [4–8] and can be functionalized for a variety of applications [9–12].

[Added:] On the other hand, signatures of ferromagnetic domains and superconductivity at the interfaces of these materials suggest the possibility for inhomogeneous distributions of the interfacial excess charge [13–18].

The following references have been added to substantiate the modified text: [13] Li et al. (2011), [14] Ariando et al. (2011), [15] Ristic et al. (2011), [16] Bucheli et al. (2015), [18] Zhang et al. (2021). Ref [17] (Bovenzi et al., 2019) was already present in the original manuscript, with a different number.

In addition, we partly-modified the following paragraph:

This [$\text{KTaO}_3(001)$] is a prototypical polar surface expected to host a homogeneous 2DEG on the ferroelectrically-polarized TaO_2 termination [27]. Our data show that the intrinsic excess charge on KTaO_3 is instead accommodated through the formation of charge-density waves (CDW, periodic charge modulations occurring in a variety of materials) [28–31] and highly-localized small electron polarons and bipolarons [...]. The formation of ordered patterns of (bi)polarons and a CDW on the ferro electrically active $\text{KTaO}_3(001)$ is expected to impact the functionalities of the material as compared to the simple 2DEG picture. From a more fundamental point of view, our study demonstrates much richer physics of polar surfaces than considered so far, revealing charge modulation as an intrinsic property of the bare surface.

Here, we added Ref. [28] (Chen et al., 2016), and replaced a previous reference by citing now an Editorial Letter published in the last few months by the same author for a special issue of modern studies on CDW, Ref. [29] (Balandin et al., 2021).

Reviewer: 2. In order to persuade readers that one should consider the CDW or polaronic effects, e.g., in carrier transport applications, the authors should first offer one or a few quantitative analysis on experimental observations to show if there are nonnegligible effects (if exist) that CDW and polaronic states have on the 2D electronic gas behaviors.

Authors: Charge modulation and charge localization are known to show a broad impact on the functionalites of the materials. Modulation of the charge in form of CDW can be indeed related to the formation of superconductive states: the added citations reported in Point 1 of this reply reference experimental and theoretical works addressing this aspect. Similarly, polarons play a primary role in many different applications, including transport, as already described in the original Introduction:

Localized polarons cause deep modification of surface properties, fundamentally different from those induced by a dispersed 2DEG, and play a key role in all applications involving charge transport and catalysis.

Moreover, based also on recent studies (see for example our work on TiO_2 , [doi: 10.1103/PhysRevLett.122.016805](https://doi.org/10.1103/PhysRevLett.122.016805)), it is reasonable to expect that polaronic states influence molecule/atom adsorption and catalysis on the surface. Indeed, recent results that we have obtained for CO molecules on the KTaO_3 not only confirm this hypothesis, but shows a strong enhancement of the molecule-substrate binding due to the surface polarity, and to the ferroelectricity reversal brought about by the bipolarons.

Nevertheless, we agree with the Reviewer that our final statement in the original abstract was too vague, thus, we have removed it from the amended version:

[Removed statement:] Controlling the degree of charge ordering and the transition from ferro electric to paraelectric states could be of great benefit for the generation and transport of carriers in electronic applications.

Reviewer: 3. The authors discussed the STM observations of CDW state and polaronic state in a clear way, however the DFT simulation results do not agree with the observations. The CDW DOS, according to the STM results, are close to the Fermi level, while in all simulated structures shown in the current work, only the pure CDW structure (ferromagnetic Fig.2a and antiferromagnetic Fig.SF6a), the bipolaron structure shown in Fig.SF9b, and the mixture structure of CDW+polaron shown in Fig.SF10b have CDW DOS nearby the Fermi level, all of the three having significant higher total energies (>0.1 eV) than the global ground state (the one shown in Fig.3b) and hence could become unstable. The ground state, on the other hand, does not host any CDW state nearby the Fermi level. Before discussing the CDW and polaronic effects based on the DFT results, the authors should first solve these disagreements.

Authors: The analysis of the DOS performed with DFT calculations for every phase separately (CDW, polarons, bipolarons, and mixed phases) allows us to confirm the interpretation of the STS spectral signal (in addition to the indications given by the spatially resolved STM maps): STS peaks deep in the gap can be attributed to polarons and bipolarons, while the shallow peak corresponds to the CDW (or to the strongly interacting bipolarons aligned along (100) directions reported in the Supplement, which, however, is quite an unlikely scenario considering the distribution of localized spots in the experimental STM maps).

The Reviewer is right noting that (bi)polarons and CDW are found to coexist in the experiment while the polaronic states are energetically more favorable in our DFT calculations. There are different aspects to consider in order to explain this apparent disagreement. We have included a discussion in the revised version of the main text:

Our computational data show that polaronic states can coexist with the CDW, in agreement with the experiment. We note that a precise characterization of the relative stability of the electronic surface states is hampered in the DFT calculations by the dependence of the free energy on the computational setup (see Fig. SF12) [42]; moreover, the infinitely large TaO₂ terminations in the calculations neglect the effects of the surface electrostatic field arising from the neighboring KO terraces, known to alter the surface electronic states [27]; also, the limited size of the DFT unit cell constrains the electronic phases, the CDW in particular. Nevertheless, the small differences in the formation energy of the different phases (~ 50 to ~ 250 meV), including the CDW+polaron phase, support the possibility of a coexistence of these states as observed on the experimental samples.

The CDW is indeed the state that is penalized to the largest extent by our DFT setup. First of all, the limited size of the unit cell constrains the CDW to a wavelength smaller than what observed in the experiments, reducing its stability.

Secondly, we note also that the energies reported for the polaronic phases are calculated considering (i) optimal patterns of the (bi)polarons (not necessarily achieved in the rather small TaO₂ terraces of the experimental samples), and (ii) optimal polaronic distortions around the localization sites: While the CDW appears resilient against lattice distortions (the $\sqrt{2} \times \sqrt{2}$ electronic modulation of the CDW is stable even in the 1×1 lattice, with the ions not matching the periodicity of the charge modulation), the stability of the (bi)polarons is strongly dependent on the local structure. Therefore, the 0 K calculations tend to favor the polaronic formation energies, neglecting any detrimental effect arising from thermal vibrations. Finally, a precise characterization of the relative stabilities is hampered in the DFT calculations by the dependence of the energy on the computational setup (different localized states are affected to different extent by the computational parameters, as recently shown in Ref. 42, see also Point 5 of this reply).

Reviewer: 4. The authors should offer enough details about how they achieved the many structures with energies between the pure 2DEG and the ground states. Are they the local minima DFT found, or from constrained-DFT optimizations?

Authors: All solutions are indeed genuine (un-constrained) DFT local minima, as briefly mentioned in the Methods. It is in fact sufficient to adopt different initial conditions (for example, breaking the lattice symmetry by locally elongating selected Ta-O bonds, or by adopting initial wave-functions different than the default ones) to drive the DFT convergence towards one local

minimum or the other. We note that, as reported in our study, the conventional solution of the homogeneous 2DEG is nothing else but a local minimum, which is typically obtained in the calculation if the lattice structure and the wavefunctions are initialized to a 1×1 symmetry (the conventional choice in all studies on polar surfaces and interfaces so far). Considering the Reviewer's comment (and a similar comment from Reviewer 3), we have now included a new section in the Supplementary Information, "Modeling the Phases of the Excess Charge", to better describe this aspect.

Reviewer: 5. The authors used the SCAN+U method with $U = 4$ eV on Ta d-orbital. The first question is how this U value is determined. Is it from constraint RPA, or linear response, or an arbitrary value? For example, the linear-response method gives smaller $U=2.27$ eV for bulk Ta in GGA+U method, while SCAN often requires a smaller U value than GGA. The authors mentioned that the CDW or polaronic state will no longer be the ground state when $U=0$. Does it mean that the conclusions of the current work, at least the theoretical part, are U -dependent and might not survive under another arbitrary U value?

Authors: We agree with the Reviewer (and Reviewer 3 as well) that the motivations for our choice on the computational setup should be reported: We have expanded the Supplementary Information with one additional section, "The U parameter", and one additional Figure, Fig. SF12, where we discuss our choice in detail.

In brief, we have identified a computational setup that enabled us to properly reproduce the band gap at the DFT+ U level and, simultaneously, agreed with the value of U obtained by cRPA. The value of $U = 4$ eV used in combination with SCAN functional in bulk KTaO_3 , allows us to reproduce the band gap as obtained by hybrid functional calculations at higher computational cost (unaffordable for the surface slab calculations). By using SCAN wave-functions, the cRPA Hubbard U for the Ta atoms in bulk KTaO_3 is $U_{\text{cRPA-SCAN}}^{t2g} = 3.25$ eV (larger than for PBE wave-functions). We note that this value represents a lower limit for the U adopted in the DFT+ U calculations, since the e_g states were not included in the cRPA calculation (because these states are entangled with other bands). Moreover, surface and near-surface atoms are affected by electronic screening to different extent as compared to the bulk. For these reasons, we considered $U = 4$ eV as an adequate choice.

We highlight that the CDW and (bi)polaronic states would exist also for different choices of U : It requires significantly small U values ($U \approx 2$), not supported by the cRPA and band gap analysis, to suppress all these electronic states. This explains the lack of such states in previous DFT studies that omitted entirely corrections of the electronic correlation (*i.e.*, purely LDA/GGA calculations).

Reply to the Third Reviewer

Reviewer: I have read with great interest the manuscript titled "Competing electronic states emerging on polar surfaces" from Reticcioli et al that discusses charge excesses at polar surfaces of oxides. While one usually assumes that a "simple" 2D electron gas forms at the surface in order to cancel/accomodate the polar catastrophe and/or the electric field at the interface with vacuum, authors here show that polaronic and bipolaronic states can form at the surface. Results are supported by first-principles simulations as well as scanning probe microscopy.

Firstly, the paper is very well written, the problem is accurately introduced and the whole story is accessible for a broad audience. Secondly, the physics and results discussed in the manuscript are important as authors show that instead of being delocalized over few unit cells below the surface (typically 1 to 3 uc) and form a 2D gas, the excess charge can be trapped and form polaronic states. To the best of my knowledge, this is a largely ignored scenario and it is thus a breakthrough as these results hint at new paradigms for the physics of surfaces.

Despite of the few comments on the manuscript appended below that have to be addressed, I strongly support publication of the manuscript in Nature Communications.

Being more an expert of first-principles simulations, I have few questions regarding the procedures and part of the results:

Authors: We thank the Reviewer for her/his positive evaluation on our study, and for supporting the publication of the manuscript. We share her/his enthusiasm on the novelty of the results, and we thank the Reviewer for the very constructive comments that have largely contributed to increase the readability and quality of the manuscript in its revised form.

Reviewer: - How do the authors reach the solution with trapped electrons? Do they perform an initial calculations with a pattern of specific orbital occupancies and then let the solver relax the total energy and forces in a second step?

Authors: Motivated by the Reviewer's comment (as well as Reviewer 2), we have now included a new section in the Supplementary Information, "Modeling the Phases of the Excess Charge", to better describe this aspect.

We did not perform any constrained calculation. The (bi)polarons were modeled with one of the standard approaches for localizing polaronic states: (i) we initially introduced local lattice distortions (elongated Ta-O bonds) around the Ta ion chosen to host a (bi)polaron, (ii) then, we initialized the local magnetic moments of these Ta ions to $\Gamma^1 1/t_B$ for polarons and $\Gamma^1 2/t_B$ for bipolarons, (iii) and, finally, we let the system converge to the structurally and electronically relaxed solution.

Regarding the CDW, the most efficient strategy turned out to use the undistorted lattice (the polarons in the $\sqrt{2} \times \sqrt{2}$ arrangement: with this approach, since it is not possible to maintain the 1×1 lattice of the 2DEG) and to initialize the wavefunctions to those obtained for the sub-surface polaronic wavefunctions (which would require *ad-hoc* lattice distortions), the calculation converges to a solution, the CDW, with different wavefunctions (compare the bands in Fig. 4b and 4c) but with the same charge periodicity. Alternatively, the CDW can also be obtained as a spontaneous result in the mixed CDW+polaron phase by progressively distorting the lattice from the 2DEG to the (bi)polaronic structures (Fig. 5 in the main text).

Reviewer: - Authors use the SCAN functional that was previously shown to be appropriate for ABO₃ oxides with a 3d element (PRB 100 035119). Nevertheless, they used SCAN with a +U potential on Ta d states. What is the reason for that choice? I would expect that delocalization errors are smaller for 5d elements than for 3d elements. Is it just a delocalization error appearing at the surface?

Authors: The Reviewer raises an interesting point. We note that the work cited by the Reviewer (PRB 100 035119) refers to magnetic transition metal perovskites with partially filled 3d states (d^1 -to- d^8), whereas KTaO₃ is a non-magnetic 5d⁰ system, thus, not falling in the category of Mott-insulators, the focus on the cited paper. The type (static vs dynamic) of correlation and its strength (weak or intermediate) in 5d systems is a rather open subject and no general rationale have been proposed so far due to the overall small number of studies dealing with this class of materials. Our experience on 5d materials suggests that the cRPA value of U is a valid choice for the description of electronic properties of 5d TMOs. In fact, the SCAN+ U setup allows us to reproduce with good accuracy the expected energy band gap of KTaO₃ (see the new Section in the Supplementary Information, "The U parameter"). In addition, the main role of U in this study is to allow for localization of the excess charge (see point below).

Reviewer: - Authors write that when removing the +U in the simulations, the polaronic states disappear. It goes along with the previous point: is it really a physical effect or a problem from the SCAN functional? I guess another parameter free simulations such as HSE06 on the smallest slab possible could provide a full justification of this point.

Authors: We thank the Reviewer for underlining this aspect (in analogy with Reviewer 2). We have included a description of the effects of U on the electronic surface state in the section already mentioned in the previous point, "The U parameter", and in the corresponding Fig. SF12. First, we underline that all states were obtained also by using the PBE+ U setup (but we consider the SCAN+ U setup more accurate). Second, in the amended version of the

Supplementary Information, we show that even small values of U (smaller than the lower limit given by cRPA, see previous point) are able to reproduce all the surface states. Conversely, by neglecting completely the correction to the electronic correlation, *i.e.*, in SCAN/PBE calculations with no U applied (the setup adopted by all studies so far), the only solution that can be achieved is the spin degenerate 2DEG.

Following the Reviewer's suggestion, we tried to adopt also the hybrid functional. We were able to obtain both polarons and bipolarons in electronically self consistent calculations, keeping the ions fixed at the SCAN+ U positions due to the high computational cost of HSE. In fact, we cannot reduce the size of the slab to facilitate the calculations: a large lateral size is required to avoid self interaction of the localized charge with its image in the periodic unit cell, and several atomic layers are required to correctly describe the ferroelectric-like distortions.

Reviewer: - Authors observe spin polarized 2DEG at the surface. Do they have an idea of the origin of such a spin polarized gas? Have they checked if this could be antiferromagnetically ordered?

Authors: This is an interesting question. As it appears from the newly added Fig. SF12, the spin polarization of the 2DEG seems to arise as a correlation effect (the 2DEG is spin degenerate for $U = 2$ eV). We did try to stabilize antiferromagnetically ordered surface structures, but unsuccessfully.

REVIEWERS' COMMENTS

Reviewer #1 (Remarks to the Author):

I am overall satisfied with the authors' reply and changes to the manuscript. Below are my further questions:

Question 1:

In their reply the authors said that they have included in Fig. SF5 in the revised SI an averaged dI/dV - V spectrum recorded in empty-state energies, as supplementary to that recorded at filled-state energies shown in Figure 1f. However, I could not find such a spectrum in Fig. SF5. Please provide such a spectrum if the authors have it.

Question 2:

I appreciate the authors for providing additional data showing the disappearance of the CDW at ~ 80 K. I have a follow-up question regarding the LDOS: in the additional text in the revised SI, page 2, the authors mentioned about the unconventional disappearance of the LDOS near the Fermi level at elevated temperatures, as evidenced by the dramatic decrease in tunneling current at comparable height, with the heights at two different temperatures (5 and 80 K) estimated from the AFM images. How dramatic is the reduction of tunneling current? 5 times? 10 times? Or even more dramatic? The reason behind my question is that tunneling current is extremely sensitive to tip-sample distance, hence even a slight change in tip-sample distance can lead to appreciable change in current. The authors ought to be more quantitative about this. As an suggestion, a direct comparison of the dI/dV spectra at different temperatures may serve as a stronger evidence for the point the authors try to establish.

So long as the authors provide satisfactory answers to my questions, I recommend that their manuscript be published as a Research Article in Nature Communications.

Reviewer #2 (Remarks to the Author):

As all comments and questions from me in the last iteration have been addressed and answered in a proper way in the current stage, I would like to support the current manuscript as a publication in Nature communications.

Reviewer #3 (Remarks to the Author):

Authors have properly addressed the comments raised at the first round. They further provide additional data supporting their conclusions. I therefore recommend the publication of the manuscript in Nature Communications.

Reply to the First Reviewer

Reviewer: I am overall satisfied with the authors' reply and changes to the manuscript. Below are my further questions:

Question 1: In their reply the authors said that they have included in Fig. SF5 in the revised SI an averaged dI/dV - V spectrum recorded in empty-state energies, as supplementary to that recorded at filled-state energies shown in Figure 1f. However, I could not find such a spectrum in Fig. SF5. Please provide such a spectrum if the authors have it.

Authors: We have the spectrum and it is now included in the Supplementary Figure 5. There is no pronounced peak structure in empty states. KTaO_3 is an n-type semiconductor, therefore the empty-states LDOS is dominated by the onset of the conduction band.

Reviewer: Question 2: I appreciate the authors for providing additional data showing the disappearance of the CDW at 80 K. I have a follow-up question regarding the LDOS: in the additional text in the revised SI, page 2, the authors mentioned about the unconventional disappearance of the LDOS near the Fermi level at elevated temperatures, as evidenced by the dramatic decrease in tunneling current at comparable height, with the heights at two different temperatures (5 and 80 K) estimated from the AFM images. How dramatic is the reduction of tunneling current? 5 times? 10 times? Or even more dramatic? The reason behind my question is that tunneling current is extremely sensitive to tip-sample distance, hence even a slight change in tip-sample distance can lead to appreciable change in current. The authors ought to be more quantitative about this. As an suggestion, a direct comparison of the dI/dV spectra at different temperatures may serve as a stronger evidence for the point the authors try to establish.

Authors: The observed change ranges from 2 to 4 orders of magnitude (100 times to 10000 times), i.e., the tunnelling current decreases down to non-measurable values. This is apparent from the scale bars in the Supplementary Figure 6, and we now also mention it in the Supplementary text. This observation comes from our systematic work on this material performed over several years. We have a large number of STM/AFM images supporting this claim, but we unfortunately do not have a systematic set of high-quality STS spectra.

Reviewer: So long as the authors provide satisfactory answers to my questions, I recommend that their manuscript be published as a Research Article in Nature Communications.

Reply to the Second Reviewer

Reviewer: As all comments and questions from me in the last iteration have been addressed and answered in a proper way in the current stage, I would like to support the current manuscript as a publication in Nature communications.

Authors: We are glad to read about the positive evaluation from the Reviewer, and we thank her/him again for the very constructive comments.

Reply to the Third Reviewer

Reviewer: Authors have properly addressed the comments raised at the first round. They further provide additional data supporting their conclusions. I therefore recommend the publication of the manuscript in Nature Communications.

Authors: We would like to thank the Reviewer for her/his positive consideration on our work, and for all the recommendation received during the review process.